



# The effect of greenhouse gas concentrations and ice sheets on the glacial AMOC in a coupled climate model

Marlene Klockmann[1,2], Uwe Mikolajewicz[1], and Jochem Marotzke[1]

[1]Max Planck Institute for Meteorology, Hamburg, Germany

[2]International Max Planck Research School on Earth System Modelling, Hamburg, Germany

*Correspondence to:* Marlene Klockmann (marlene.klockmann@mpimet.mpg.de)

**Abstract.** Simulations with the Max Planck Institute Earth System Model (MPI-ESM) are used to study the sensitivity of the AMOC and the deep ocean water masses during the Last Glacial Maximum to different sets of forcings. Analysing the individual contributions of the glacial forcings reveals that the ice sheets cause an increase in the overturning strength and a deepening of the North Atlantic Deep Water (NADW) cell, while the low greenhouse gas (GHG) concentrations cause a

decrease in overturning strength and a shoaling of the NADW cell. The effect of the orbital configuration is negligible. The effects of the ice sheets and the GHG reduction balance each other in the deep ocean so that no shoaling of the NADW cell is simulated in the full glacial state. Experiments in which different GHG concentrations with linearly decreasing radiative forcing are applied to a setup with glacial ice sheets and orbital configuration show that GHG concentrations below the glacial level are necessary to cause a shoaling of the NADW cell with respect to the preindustrial state in MPI-ESM. For a $pCO_2$ of

149 ppm, the simulated overturning state and the deep ocean water masses are in best agreement with the glacial state inferred from proxy data. Sensitivity studies confirm that brine release and shelf convection in the Southern Ocean are key processes for the shoaling of the NADW cell. Shoaling occurs only when Southern Ocean shelf water contributes significantly to the formation of Antarctic Bottom Water.

**Key words:** AMOC, Last Glacial Maximum, climate modelling, glacial forcing

## 1 Introduction

The Last Glacial Maximum (LGM, 21 ky b.p.) provides an important test case for numerical climate models because of its low greenhouse gas (GHG) concentrations, vast ice sheets and different orbital configuration. State-of-the-art climate models succeed in simulating the mean glacial surface climate in reasonable agreement with reconstructions from proxy data. The

response of the Atlantic Meridional Overturning Circulation (AMOC) and the deep Atlantic water masses to the glacial forcing, however, often differ strongly between models and reconstructions and also between different models. The cause of these



differences is not yet well understood. Therefore we identify and analyse here the individual effects of the glacial forcings on the glacial AMOC and the glacial deep water masses.

The two overturning cells of the AMOC are tightly connected to the two water masses which dominate the Atlantic below 1000 m. The upper overturning cell is associated with North Atlantic Deep Water (NADW), the lower overturning cell is

associated with Antarctic Bottom Water (AABW). Reconstructions based on $\delta^{13}$C suggested that the boundary between these two water masses moved upwards during the LGM in both the eastern (Duplessy et al., 1988) and the western (Curry and Oppo, 2005) Atlantic basin. This indicates that NADW occupied a shallower layer and that AABW penetrated much further into the North Atlantic than today. Recently, a combination of $\delta^{13}$C, Cd/Ca and $\delta^{18}$O measurements with a tracer transport model confirmed that the core of NADW was shifted upwards during the LGM (Gebbie, 2014).

The shoaling of the boundary between NADW and AABW inferred from the $\delta^{13}$C measurements was initially interpreted as a shoaling and weakening of the upper overturning cell of the AMOC which is associated with NADW. A carbon-cycle model coupled to an ocean model could best reproduce the $\delta^{13}$C distribution over most of the Atlantic with a shallower and weaker AMOC (Winguth et al., 1999). While there is little debate about the shoaling nowadays, it remains difficult to constrain the strength of the NADW cell from reconstructions. Reconstructions based on the $^{231}$Pa/$^{230}$Th ratio (hereafter referred to

as Pa/Th) indicated that the AMOC may have been 30 to 40 % weaker during the LGM than today (McManus et al., 2004). However, the uncertainties of the Pa/Th are still quite large and a variety of possible circulation states could be made consistent with the Pa/Th measurements, depending on the assumptions made about particle fluxes and scavenging rates (Burke et al., 2011). A glacial NADW cell which was at least as strong as or even stronger than today was indicated by Pa/Th measurements in combination with a two-dimensional scavenging model (Lippold et al., 2012) and $\varepsilon$Nd measurements (Böhm et al., 2015).

In the second phase of the Paleoclimate Modelling Intercomparison Project (PMIP2), five out of nine models simulated a shallower and weaker NADW cell, while four models simulated a deeper and stronger NADW cell (Weber et al., 2007). This inter-model spread was reduced in PMIP3 with most models simulating an increase in NADW strength and a deepening of the NADW cell (Muglia and Schmittner, 2015). The large inter-model spread in PMIP2 and the fact that most models fail to simulate the shoaling of the NADW cell in PMIP3 suggests that there is still a lack of understanding of the mechanisms that

determine the glacial AMOC state.

Identifying the individual effects of the glacial orbit, ice sheets and GHG concentrations on the climate can help to understand the full response. Many studies have addressed the effects of the individual glacial forcings on the atmosphere and the surface ocean using either an atmospheric general circulation model coupled to a mixed-layer ocean model (e.g., Broccoli and Manabe, 1987; Hewitt and Mitchell, 1997; Felzer et al., 1998) or a fully coupled atmosphere-ocean model (Justino et al., 2005; Pausata et al.,

2011). Very few studies have addressed the effects of the glacial forcings on the AMOC in a coupled model framework (Kim, 2004; Brady et al., 2013) but none of them has analysed the individual effects of all three glacial forcings and none of them addressed the effects on the properties and distribution of NADW and AABW. Oka et al. (2012) studied the effects of the individual glacial surface fluxes of heat, freshwater and momentum on the state of the glacial AMOC in a stand-alone ocean model. They suggested that a thermal threshold determined the strength of the NADW cell. The location of this threshold depended on

the wind stress: if the glacial wind stress was applied, stronger cooling was necessary to cross the threshold than if the prein-





dustrial wind stress was used. It is, however, not certain that a similar threshold would be found in a coupled climate model (Marotzke, 2012). Here we study the response of the glacial AMOC and the deep water masses to different sets of forcings in the Max Planck Institute Earth System Model (MPI-ESM). Our set of simulations allows us to study the full glacial response, to separate the impact of the individual glacial forcings, and to analyse the sensitivity of the glacial AMOC to different GHG

concentrations. The last point also aims at exploring whether a similar threshold as the one identified by Oka et al. (2012) can be found in a coupled climate model.

The paper is organised as follows. The model and the experiments are described in Sect. 2 and 3, respectively. We then begin our analyses by describing the simulated state of the present-day and glacial AMOC as well as the response of the surface climate and the deep ocean to the full glacial forcing in Sect. 4. The respective impacts of orbit, GHG concentrations and

ice sheets are discussed in Sect. 5. In Sect. 6, we analyse the effect of different GHG concentrations with linearly decreasing radiative forcing on the AMOC and the deep water mass properties. The results are compared with previous studies and discussed in the respective sections where appropriate. We discuss the major findings and implications in Sect. 7 and present our conclusions in Sect. 8.

## 2   Model description

We use the physical part of the Max Planck Institute Earth System Model (MPI-ESM) in coarse-resolution mode. The model version is very similar to the CMIP5 version (Giorgetta et al., 2013). The atmosphere component is ECHAM6.1 with a spectral core in T31 which gives a horizontal resolution of approximately 3.75° x 3.75° in grid space (see Stevens et al., 2013, for details). There are 31 vertical $\sigma$-hybrid layers. The land-surface model is JSBACH including natural dynamic vegetation (Reick et al., 2013). The ocean component is MPIOM (Marsland et al., 2003; Jungclaus et al., 2006). It is a free-surface

primitive-equation ocean model on z-coordinates with an incorporated Hibler-type sea-ice model (see Marsland et al., 2003; Notz et al., 2013, for details)

MPIOM is run on a curvilinear grid with a nominal resolution of 3° x 3°. The grid poles are located over Greenland and Antarctica. This configuration produces a minimum grid spacing of 31 km around Greenland and 86 km around Antarctica. The maximum grid spacing is 285 km in the tropical Atlantic and 390 km in the tropical Pacific. There are 40 unevenly spaced

vertical levels. The uppermost layer has a thickness of 15 m in order to avoid problems with thick sea ice in the glacial Arctic Ocean. Below the first level, the level thickness increases monotonously from 10 m close the surface to about 550 m in the deep ocean. The upper 100 m of the water column are represented by nine levels. Partial grid cells fully resolve the bottom topography. The representation of shelf convection and flow over sills is improved by a slope convection scheme described in Marsland et al. (2003).

ECHAM and MPIOM are coupled daily using the Ocean Atmosphere Sea Ice Soil (OASIS3-MCT, Valcke (2013)) coupler.

The model configuration applied here is very similar to the model version MPI-ESM-P, which can be found in the CMIP5 database and which also participated in PMIP3. The characteristics of the same MPIOM version with higher resolution (MPI-ESM-LR and MPI-ESM-MR) are described by Jungclaus et al. (2013) in greater detail.



## 3  Experiments

We perform two reference simulations, a preindustrial control run (hereafter referred to as piCTL) with preindustrial GHG concentrations, present-day orbit, land-sea mask, topography and ice sheets and a PMIP3-like LGM run (hereafter referred to as LGM-ref) with glacial GHG concentrations as well as glacial orbit, land-sea mask, topography and ice sheets (see Table 1).

The glacial ocean bathymetry and land-sea mask are obtained by adding the anomalies from the ICE-5G reconstructions (0k-21k, Peltier (2004)) to the present-day bathymetry. This results in a global mean sea-level drop of approximately 124 m. The continental ice sheets correspond to the PMIP3 boundary conditions. They are a blended product of the three ice-sheet reconstructions ICE-6G (Peltier et al., 2015; Argus et al., 2014), MOCA (Tarasov et al., 2012) and ANU (Lambeck et al., 2010) (see Abe-Ouchi et al. (2015) and the PMIP3 web page for a more detailed description: https://pmip3.lsce.ipsl.fr).

We perform a second control simulation which has the same configuration as piCTL but with the orbital parameters set to glacial values (see Table 2). This simulation will be referred to as piTOPO. The difference between piTOPO and piCTL yields the response to the orbital forcing. We will use piCTL as a reference to estimate the total effect of all glacial forcings and piTOPO whenever we want to ensure that the orbital effect configuration is excluded.

To analyse the effect of the ice sheets and GHG concentrations, we perform an experiment with glacial orbit and ice sheets but

with preindustrial GHG concentrations (LGM-284). Comparing LGM-284 with piTOPO gives the response to the combination of the glacial ice sheets, glacial topography and the glacial land-sea mask. In the following, we will simply refer to this combined response as the ice-sheet effect. Comparing LGM-ref with LGM-284 gives the response to the GHG reduction.

To analyse the effect of the GHG reduction on the glacial AMOC in detail, we apply different GHG concentrations to a setup with glacial orbit, topography, ice sheets and land-sea mask. The GHG concentrations are chosen such that they cover a

wide range of the parameter space. To determine the interval spacing, we calculate the difference in radiative forcing (calculated according to Myhre et al. (1998)) between the preindustrial and glacial GHG concentrations and then take $\Delta RF = \frac{1}{2}(RF_{piCTL} - RF_{LGM}) = 1.45$ Wm$^{-2}$ as the interval spacing, so that the radiative forcing decreases approximately linearly between the experiments. We change the concentrations of $CO_2$, $N_2O$ and $CH_4$ but will in the following refer only to $pCO_2$ for simplicity. $pCO_2$ ranges from 353 ppm to 149 ppm. In analogy to LGM-284, these experiments are named LGM-*nnn*, *nnn* being the $CO_2$

concentration in ppm.

We perform three sensitivity experiments to assess the importance of brine release in the Southern Ocean for the state of the overturning. In these experiments, we reduce the amount of brine which is released during sea-ice formation on the Southern Hemisphere by approximately 50 % (see Table 1 and Sect.6.5 for details).

Table 1 provides a complete list of the experiments and their respective setups. All experiments are integrated for at least

1400 years to reach a quasi-equilibrium. This approach is justified because the forcing at the LGM was relatively stationary over a few millennia. In the following, we analyse averages of the last 300 years of the respective simulations if not stated otherwise.



## 4 The control and glacial climates

In this section we describe the response of the AMOC and the mean climate to the combined glacial forcings. We begin with the response of the surface climate, because the surface conditions are important for water-mass formation. We then continue with the resulting changes of the water masses in the deep Atlantic and the associated AMOC response. To evaluate how well

MPI-ESM is simulating the overturning, we discuss here not only the response of the AMOC to the glacial forcing but also the AMOC state simulated in piCTL. We further compare our results with reconstructions and with previous results from PMIP2, PMIP3 and other simulations.

### 4.1 Surface air temperature

The simulated global-mean surface air temperature in LGM-ref cools by 5.18°C with respect to piCTL. The most recent

estimate of the global-mean cooling based on reconstructions is $4.0 \pm 0.8$°C (Annan and Hargreaves, 2013; Shakun et al., 2012). In PMIP2, the global-mean cooling ranged from 3.6 to 5.7°C (Braconnot et al., 2007) and five PMIP3 models evaluated by Braconnot and Kageyama (2015) simulated a global-mean cooling ranging from 4.41 to 5°C. Hence, our simulated estimate appears reasonable, being slightly colder than the reconstructions and well within the range of previous simulations.

The strongest cooling in the LGM-ref simulation takes place over the ice sheets in response to the ice-sheet elevation and

albedo (Fig. 1a). Cooling over the ice-free continents ranges from 3 to 8°C. A weak warming can be seen over the North Atlantic and over the Gulf of Alaska. The warming over the North Atlantic is an imprint of underlying warm SSTs (see Sect. 4.2), the warming over the Gulf of Alaska is generally explained by atmospheric circulation changes due to the Laurentide ice sheet (see e.g., Justino et al. (2005), Otto-Bliesner et al. (2006) and Sect. 5).

The simulated latitudinal cooling pattern agrees well with reconstructions between 60°S and 60°N, as a comparison of simu-

lated and reconstructed surface air temperature differences over land at the proxy sites shows (Fig. 2a). The reconstructions are taken from Bartlein et al. (2011) and Shakun et al. (2012). The simulated temperatures have been height corrected to account for discrepancies between the coarse model topography and the actual high-resolution PMIP3 topography. Still, the model overestimates cooling over Antarctica and in the vicinity of the ice sheets north of 60°N. These areas are characterised by large height gradients, which may be resolved neither by the coarse model grid nor the higher-resolution PMIP3 topography. This

issue was recognised as 'representativeness error' by Hargreaves et al. (2013). In addition, the surface temperature over glaciers cannot exceed 0°C within the model which may induce additional cooling directly over and downstream of the glaciers. This may explain in part why our global-mean estimate of the surface cooling is slightly stronger than the reconstructed estimate of Annan and Hargreaves (2013).

### 4.2 Sea Surface Temperature

The simulated global-mean SST in LGM-ref cools by 2.61°C with respect to piCTL. This fits within the range of the MARGO reconstructions which indicate a cooling of $1.9 \pm 1.8$°C (MARGO Project Members, 2009). It is also similar to previous results from Brady et al. (2013), who simulated a $\Delta$SST of 2.4°C with CCSM4.



The tropical ocean cools by 1.6 to 2.6°C (Fig. 3a), which is also in good agreement with the MARGO estimate of $1.7 \pm 1$°C and the range of the PMIP2 ensemble of 1 to 2.4°C (Otto-Bliesner et al., 2009). The cooling increases towards subpolar latitudes where it exceeds 4°C in the Labrador Sea, the Nordic Seas, in the North Pacific and over the Antarctic Circumpolar Current. The Arctic Ocean does not show any significant cooling as the Arctic surface waters are already close to the freezing point in piCTL. The surface ocean around Antarctica cools by 0.4 to 0.8°C. The central North Atlantic warms by 1.6 to 2.6°C. This warming is caused by a shift in the subtropical-subpolar gyre system (Fig. 4a), which enhances the transport of warm subtropical water to the North Atlantic.

Comparing the simulated SST differences with the MARGO reconstructions at the proxy sites as a function of latitude (Fig. 2b), shows that the model is always within the range of the reconstructions but generally colder than the proxy mean cooling. From 70°S to 30°N both model and proxies show a relatively small scatter, and the latitudinal pattern of cooling is quite similar. North of 30°N, the scatter of both model and proxies increases, and there is very little agreement between the two. Although the North Atlantic and Nordic Seas are the most densely sampled areas, it remains very difficult to constrain the temperature anomaly in this region due to divergent proxy results (MARGO Project Members, 2009). In fact, the simulated SST differences agree quite well with the reconstructions based on foraminifera, but there is little agreement with reconstructions based on dinoflagellates and alkenones.

### 4.3 Sea Surface Salinity

The lower sea level leads to a global-mean salinity increase of about $1.21 \, \mathrm{g \, kg^{-1}}$. This corresponds to the increase seen over most of the tropical ocean, where salinity increases by 0.5 to $1.5 \, \mathrm{g \, kg^{-1}}$ (Fig. 3c). Larger increases can be seen on the shelves of the Weddell Sea, Baffin Bay and Beaufort Sea. In the Mediterranean the salinity increase exceeds $5 \, \mathrm{g \, kg^{-1}}$ due to the reduced exchange with the Atlantic (Mikolajewicz, 2011). The eastern North Atlantic shows a salinity increase of about $2.5 \, \mathrm{g \, kg^{-1}}$ which can also be attributed to the shift in the subtropical-subpolar gyre system (Fig. 4a). Freshening occurs in areas where runoff from the Laurentide and Fennoscandian ice sheets reaches the ocean.

### 4.4 Deep water masses in the Atlantic

A north-south section of temperature differences between LGM-ref and piCTL through the Atlantic shows that cooling occurs over the entire water column (Fig. 5b). The cooling is strongest at about 2500 m depth, right above the boundary between NADW and AABW in piCTL. We infer the water-mass boundary from the location of the strongest vertical gradient in both temperature and salinity (Fig. 5a and e). There is, however, no clear indication of an upward shift of the water-mass boundary in LGM-ref.

Also the salinity anomalies do not indicate an upwards shift of the water-mass boundary (Fig. 5f). Instead, they show that the salinity difference between AABW and NADW decreases. The salinity increase is strongest in the deep Weddell Sea. Here, the salinity increase is on the order of $0.3 \, \mathrm{g \, kg^{-1}}$ above the global-mean increase. In contrast, the salinity increase of NADW is smaller than the global mean increase. Because the salinity increase of AABW is larger than that of NADW, the north-south salinity difference in the deep Atlantic is reduced by a factor of three to $0.1 \, \mathrm{g \, kg^{-1}}$. The simulated salinity increase of AABW





is weaker than suggested by reconstructions. Adkins et al. (2002) found that the north-south salinity gradient in the deep glacial Atlantic was reversed with respect to today, with AABW being saltier than NADW. They found that glacial AABW was about $2.4 \pm 0.17\,\mathrm{g\,kg^{-1}}$ saltier than the preindustrial AABW. The simulated salinity difference between LGM-ref and piCTL in the Weddell Sea is about $1.5\,\mathrm{g\,kg^{-1}}$. Therefore, the simulated glacial AABW is not salty enough to produce a north-south salinity

gradient of the correct sign and magnitude. This is a problem that many coupled LGM simulations have in common: out of the nine models that participated in PMIP2, only one succeeded in producing a salinity increase in the deep Southern Ocean of a comparable magnitude (Weber et al., 2007), and only two were able to simulate the reversal of the north-south salinity gradient (Otto-Bliesner et al., 2007).

### 4.5 Overturning

The simulated preindustrial AMOC has a maximum overturning strength of 16.5 Sv (Fig. 6a). This maximum occurs at 30°N. The northward transport of AABW has a maximum of 4 Sv near 15°N. Latest results from the RAPID-MOCHA array reveal a mean AMOC strength of $17.2 \pm 0.9\,\mathrm{Sv}$ (McCarthy et al., 2015). Estimates of northward AABW transport are within the range of 1.9 to 4 Sv (Frajka-Williams et al., 2011). Therefore, the simulated strength of the two cells lies well within the uncertainty range of the observations.

The NADW cell extends down to about 2900 m in piCTL. The boundary between the two overturning cells is quite flat, having the same depth at all latitudes. Thus, the model produces a too shallow NADW cell north of 26 °N compared with a depth of 4300 m at 26°N in the RAPID-MOCHA observations (Msadek et al., 2013). A too shallow NADW cell has also been reported for many of the preindustrial simulations in PMIP2 (Weber et al., 2007) and newer simulations (e.g., Msadek et al., 2013; Brady et al., 2013, for CCSM4).

The simulated glacial AMOC of 20 Sv exceeds the preindustrial AMOC (Fig. 6b). The depth of the NADW cell remains unchanged, as does the strength of the AABW cell. This response is quite common for models that have participated in PMIP3. All models simulate a stronger AMOC and all models but one simulate either a deepening or no change of the NADW cell depth (Muglia and Schmittner, 2015).

The relationship between the geometry of the two overturning cells and the actual vertical distribution of NADW and AABW

is not necessarily straightforward. The temperature and salinity sections in piCTL (Fig. 5a and e) show that a significant amount of NADW reaches levels below 3000 m, even though this is not depicted by the zonally integrated overturning stream function (Fig. 6a). Hence, changes in the relationship between NADW and AABW inferred only from the overturning stream function need to be interpreted with great care. In the LGM-ref simulation, however, the responses of hydrography and overturning stream function appear consistent; neither of them indicate a change in the vertical extent of NADW.





## 5 Impact of individual glacial forcings

In this section, we decompose the response to the total glacial forcing in LGM-ref into the individual contributions of orbital parameters, GHG concentrations and ice sheets (combined effect of topography, albedo and coast lines). We begin again with changes of the surface climate and then discuss the effect on the deep water masses and the resulting AMOC changes.

### 5.1 Surface temperature

The orbital configuration has the smallest effect on the annual average surface temperature distribution, a finding that is in agreement with previous studies (e.g., Hewitt and Mitchell, 1997). The tropical-temperature change is mostly smaller than ± 0.5°C (Fig. 1b). The cooling is stronger at high latitudes with 1 to 2°C. The strongest cooling is located over the Weddell Sea and the Barents Sea. In these regions, the temperature change is amplified by an expansion of the sea ice and the subsequent reduction of oceanic heat loss to the atmosphere. Because the orbital effect on the mean climate is very small, we will in the following focus only on the effects of GHG reduction and ice sheets.

The effect of the GHG reduction shows the typical pattern of a GHG reduction experiment, with polar amplification and stronger cooling over the continents than over the ocean (Fig. 1c). Also here, the strongest cooling takes place over the Weddell Sea due to the expansion of the winter sea ice. The GHG reduction accounts for most of the cooling over the ocean in the total response.

The ice sheets induce stronger cooling over the continents than over the ocean (Fig. 1d). The cooling is strongest directly over the Laurentide, Fennoscandian and West Antarctic ice sheets, from the combined effect of albedo, elevation and glacier mask (the surface temperature on the glaciers cannot be warmer than 0°C within the model). The warming over the North Pacific and North Atlantic seen in the total glacial response can be attributed to the effect of the ice sheets: There is a strong warming over the North Pacific and North Atlantic, which is also present in the SST pattern (see Fig. 3b). These warmer patches are most likely caused by circulation changes of both ocean and atmosphere. The North Pacific warming was also found in earlier modelling studies in response to both the ICE-4G (Justino et al., 2005; Kim, 2004) and the ICE-5G reconstruction (Otto-Bliesner et al., 2006). Justino et al. (2005) connected the warming to topographic blocking upstream of the Laurentide ice sheet. The warming in the North Atlantic is caused by a shift in the subtropical-subpolar gyre system in response to wind-stress changes due to the ice sheets. In LGM-284, the subtropical gyre extends further north than in piTOPO (see Fig. 4b). The maximum warming collocates with the northward extension of the subtropical gyre. Because the subtropical-subpolar gyre system is strongly controlled by the surface wind stress forcing, this warming pattern is very sensitive to the prescribed ice sheets. The warming did not occur in simulations using the ICE-4G ice sheets (Justino et al., 2005; Kim, 2004) but it was present in simulations using the ICE-5G ice sheets (Pausata et al., 2011). Ziemen et al. (2014) found that different ice-sheet configurations had a large impact on deep-water formation patterns in the North Atlantic and thus also on regional heat budgets and surface temperatures.



## 5.2 Surface salinity

The GHG reduction causes a salinity increase in high latitudes and a freshening in the entire North Atlantic and the subtropical latitudes of the Southern Hemisphere (not shown). This pattern corresponds to a weaker water cycle in cold climates. It also agrees well with the GHG effect on surface salinity found by Kim (2004). The salinity increase in the high-latitude Southern Ocean favours the formation of AABW.

The ice-sheet effect dominates the total surface salinity response (compare Fig. 3c and d). The freshening in the high northern latitudes occurs due to precipitation changes in the vicinity of the ice sheets. The strong salinity increase in the eastern North Atlantic is caused by a combination of effects: In piTOPO, the relatively fresh subpolar gyre extends very far eastwards (see contours in Fig. 4b). This causes the upper 200 m of the water column in the eastern North Atlantic to be much fresher than the underlying water. In LGM-284, the subpolar gyre retreats westwards and the subtropical gyre extends further north (Fig. 4b), enhancing the surface salinity in the eastern North Atlantic. In addition, the wind-stress anomaly due to the ice sheets (not shown) induces an offshore Ekman transport and upwelling off the Bay of Biscay and the Irish coast. This upwelling brings the saltier water from deeper layers to the surface, thus enhancing surface salinity further. Kim (2004) did not find this salinity increase in the North Atlantic in response to the ice sheets. Instead, he found that the ice sheets induced a substantial freshening in the North Atlantic (see his ICEAN effect). The salinity of the North Atlantic is an important factor for the formation of NADW, a salinity increase favours NADW formation while a salinity decrease counteracts it. Hence, the surface salinity response has a direct impact on the deep water masses and the overturning, as the next sections will show.

## 5.3 Deep water masses in the Atlantic

The ice sheets induce a warming throughout the Atlantic below 3000 m, which increases towards the north (Fig. 5d). This warming indicates a larger percentage of relatively warm NADW below 3000 m and a corresponding reduction of cold AABW. Between 1500 and 2500 m, there is a cooling associated with a weaker Mediterranean Outflow and increased convection in the Labrador Sea. In piCTL, there is a strong vertical temperature gradient centred around 3000 m, indicating the boundary between AABW and NADW, In LGM-284, this gradient is weakened by a factor of two due to the larger percentage of NADW present below 3000 m. The salinity anomalies (Fig. 5h) are similar to the temperature anomalies. There is a freshening above 2500 m in response to the reduced Mediterranean outflow. The strong salinity increase in the North Atlantic below 3000 m north of 47°N also indicates the increased fraction of NADW, which agrees well with the surface salinity increase in the North Atlantic.

The GHG reduction causes a cooling of the entire water column (Fig. 5c). The cooling is strongest north of 30°N below 3000 m, indicating a larger percentage of cold AABW and a smaller percentage of NADW below 3000 m. The surface cooling in the North Atlantic is outweighed by the surface freshening, and NADW formation is reduced. The section of salinity anomalies shows that NADW becomes fresher and AABW saltier (Fig. 5g). The strongest salinity increase takes place in the Weddell Sea both at the surface close to the coast and in the deep Weddell Sea. This increase is caused by changes in the haline density flux due to sea-ice expansion and increased brine release (see also Sect. 6.4 and 6.5).



The decreased fraction of NADW below 3000 m in response to the GHG reduction is similar to the response that the reconstructions based on $\delta^{13}C$ suggest (Duplessy et al., 1988; Curry and Oppo, 2005). However, the effect of the GHG reduction and that of the ice sheets compensate each other in the deep Atlantic, so that no clear indication of a shoaling of the water mass boundary can be observed in the LGM-ref simulation.

## 5.4 Overturning

The presence of the glacial ice sheets causes the strength of the NADW cell to increase by 7.5 Sv at 30°N (compare orange and black solid lines in Fig. 7). The boundary between the two cells (indicated by the level of zero transport) is shifted downwards by about 300 m. This reflects the increased NADW formation due to the increased northward salt transport (Fig. 3d) and is consistent with the increased fraction of NADW below 3000 m seen in the hydrographic sections (Fig. 5d and h).

The GHG reduction induces a decrease in the NADW cell strength by 4 Sv and a shoaling of the NADW cell by about 300 m (compare orange and cyan lines in Fig. 7). Again, the overturning response is in agreement with the response of the deep water masses. The GHG-induced shoaling of the NADW cell is exactly compensated by the ice-sheet-induced deepening.

While the ice-sheet effect described in previous studies varies both in sign and magnitude, the GHG effect appears to be more consistent across the different studies, at least in its sign. The ice-sheet effect depends on the ice-sheet reconstruction and also on the model. The ICE-4G ice sheets induced a weakening of the NADW cell and an expansion of the AABW cell (Kim, 2004). The PMIP3 ice sheets induced a strengthening and shoaling of the NADW cell in CCSM4 (compare experiments LGM and LGMCO$_2$ in Brady et al., 2013). In a recent study, Muglia and Schmittner (2015) found that applying glacial wind stress anomalies from the PMIP3 ensemble in the UVic model led to an increased northward salt transport, enhanced overturning and a deeper NADW cell, which is consistent with the ice-sheet effect in MPI-ESM. The sign of the GHG effect appears more robust. Both Kim (2004) and Brady et al. (2013) find a shoaling and weakening of the NADW cell as well as an enhanced AABW cell in response to the GHG reduction, which is consistent with the GHG effect in MPI-ESM. To understand the magnitude of the GHG effect on the AMOC better, we will in the following explore the impact of different GHG concentrations on the climate and consequently on the AMOC itself in the glacial setup (see the LGM-*nnn* experiments in Table 1). The aim is to explore the sensitivity of the AMOC to different GHG concentrations and to get a better understanding of the processes which determine the geometry and strength of the overturning.

# 6 Impact of different GHG concentrations

## 6.1 Overturning

Applying the different GHG concentrations to the glacial setup shows that the AMOC response to a GHG reduction is a function of the GHG concentration itself (Fig. 7). The AMOC profiles of LGM-353 and LGM-284 at 30°N are indistinguishable from each other, both in terms of NADW cell strength and depth. For $pCO_2$ below 284 ppm, the overturning decreases approximately linearly with the decreasing radiative forcing in steps of about 2 Sv per $\Delta RF$. The shoaling of the NADW cell



sets in only for $pCO_2$ below 230 ppm. LGM-353, LGM-284 and LGM-230 all have the same overturning geometry with a cell boundary near 3200 m. In LGM-ref, the cell boundary is then located at 2900 m, and in LGM-149 it shifts further upward to 2600 m. LGM-149 is the only experiment in which the NADW cell becomes shallower than in piCTL, therefore it has an overturning geometry that is in better agreement with reconstructions than that in LGM-ref. The reason for this will be explored

in the remainder of the section.

## 6.2   NADW formation

In both piCTL and piTOPO, NADW formation through deep convection takes place mainly in the ice-free part of the Nordic Seas (Fig. 8a, only piCTL is shown). The deep convection in the Labrador sea is not well captured; it takes place only sporadically, and therefore the long-term mean of the winter mixed-layer depth (MLD) is rather shallow with 400 m to 600 m. This is

a known effect of the coarse resolution. Higher-resolution versions of MPI-ESM show MLDs down to 3000 m in the Labrador Sea (Jungclaus et al., 2013).

In the relatively warm glacial experiments LGM-353 and LGM-284, the deepest mixed layers are found in the Labrador Sea (Fig. 8b and c). There is also deep convection in the Nordic Seas, but the extent of the convection area is reduced in comparison to piCTL, because sea-ice cover in the Nordic Seas increases in the glacial setup. With decreasing $pCO_2$, the sea-

ice edge advance southwards in the Nordic Seas both in summer and winter, and deep convection decreases. In the Labrador Sea, the sea-ice edge advances eastwards and the deep-water formation area shifts with it. In LGM-149, the main deep-water formation area is located in the central subpolar gyre and the eastern North Atlantic (Fig. 8f).

## 6.3   Water mass properties

To understand how the deep-water formation drives the overturning it is not sufficient to look only at the Northern Hemisphere.

The overturning strength and geometry are set to a large extent by the density difference between NADW and AABW. In the following, we analyse how the properties of the two water masses change with decreasing $pCO_2$. We choose the Nordic Seas and the North Atlantic to study the NADW properties and the Weddell Sea to study the AABW properties. In each region, we select a fixed depth representative of the water-mass properties in this region. In the Nordic Seas, we determine the water-mass properties at 560 m depth because this depth corresponds to the last wet layer in the glacial Iceland-Scotland-Channel

in MPIOM. In the North Atlantic and the Weddell Sea, we determine the water-mass properties at 2000 m depth because this depth is representative of the core properties of NADW. We then determine the spatial maximum of the climatological mean in-situ density on the selected level within each region (Fig. 9a and b) and compare the corresponding temperature and salinity (Fig. 9c). Since we compare the density from water at different depths, we converted in-situ density to potential surface density ($\sigma_\Theta$) and potential density referenced to 2000 m ($\sigma_2$) for the comparison.

In the North Atlantic, density ($\sigma_\Theta$ and $\sigma_2$) increases with decreasing $pCO_2$ (circles in Fig. 9a and b). The density increase is caused by cooling, which dominates the effect of a simultaneous freshening (circles in Fig. 9c). $\sigma_2$ increases quasi-linearly with decreasing radiative forcing; $\sigma_\Theta$ increases less below a $pCO_2$ of 230 ppm because the relative importance of the freshening effect on surface density increases at lower water temperatures.





In the Nordic Seas, both $\sigma_\Theta$ and $\sigma_2$ initially increase with decreasing $pCO_2$ until 230 ppm; at lower $pCO_2$ both decrease (diamonds in Fig. 9a and b). The initial increase is caused by cooling, the subsequent decrease by freshening and warming (diamonds in Fig. 9c). From 353 to 230 ppm, there is deep convection in the Nordic Seas associated with a strong surface-density gain and heat loss. The water that is leaving the Nordic Seas is contributing to NADW formation, because it is much

denser than that south of the Greenland-Scotland ridge. Below 230 ppm, however, the Nordic Seas are completely ice covered during winter and only little or no deep convection occurs in LGM-ref and LGM-149, respectively. The water that enters the Nordic Seas is merely recirculated without gaining density and the water that flows out over the Greenland-Scotland ridge is too light to contribute to NADW formation.

In the Weddell Sea, $\sigma_\Theta$ and $\sigma_2$ increase with decreasing $pCO_2$ (triangles in Fig. 9a and b). From 353 to 230 ppm, the density

increase is dominated by cooling (triangles in Fig. 9c). The salinity changes are positive but relatively small. Below 230 ppm, the density increase is dominated by a strong salinity increase. The cooling weakens as the Weddell Sea temperature approaches the freezing point. Even though the dominant process switches from cooling to salinity increase, $\sigma_2$ increases quasi-linearly with decreasing radiative forcing. The increase of $\sigma_\Theta$, on the other hand, strengthens below 230 ppm.

The salinity increase in the Weddell Sea and the freshening of the North Atlantic lead to a reduction and ultimately to a

sign reversal of the north-south salinity gradient, with AABW becoming saltier than NADW in LGM-149. Hence, we conclude that the simulated state of the overturning and the Southern Ocean water masses in LGM-149 are closer to the glacial state described by proxies than the state simulated in LGM-ref: The NADW cell becomes significantly shallower as compared with piCTL, and the North-South salinity gradient in the deep Atlantic is reversed with respect to the present day. This underlines the key role of the Southern Ocean salinity for the glacial AMOC state.

**6.4   The surface density flux in the Weddell Sea**

To understand the salinity increase in the Southern Ocean, we analyse the different components of the surface density flux in the Weddell Sea (defined as the region between 0 to 60°W and 60 to 90°S). We calculate the surface density flux as the sum of the density changes due to heat fluxes, freshwater fluxes (precipitation - evaporation + runoff) and brine release. We integrate these components over two different regions, the Weddell Sea shelves and the area in which open-ocean convection occurs.

This way, we can estimate the relative importance of shelf convection and open-ocean convection for the formation of AABW.

We define the shelf region as the area between the coast and the 1000 m isobath. Defining the open-ocean convection area is less straightforward because the sign and magnitude of the density flux over the open-ocean convection area is very sensitive to the definition of the latter. The most appropriate approach in terms of the flux balance would be to consider only the area in which deep convection occurs (e.g., defined by MLD exceeding 2000 m). But the integrated fluxes would be difficult to

compare this way, because the extent of the deep-convection area varies strongly between the respective experiments. We therefore choose two different approaches, each with a fixed definition of the open-ocean convection area. The first approach considers every grid point in which the annual-maximum MLD in any of the experiments exceeds 2000 m ($OOC_{max}$). The second approach considers only those grid points in which the annual-maximum MLD in all experiments exceeds 2000 m



($OOC_{min}$). The $OOC_{max}$ case is more representative of the cold experiments with a very extensive open-ocean convection area, the $OOC_{min}$ is more representative of the warm experiments with a smaller open-ocean convection area.

Over the shelves, there is a net density gain in all experiments(Fig. 10a). The gain is determined entirely by the balance of the brine component and the freshwater component. The heat-flux component is very small because the surface water is close

to the freezing point and therefore further heat loss leads to sea-ice formation and contributes to the brine component. The brine release causes a density gain which remains approximately constant with decreasing $pCO_2$ except for a stronger increase in LGM-149. The freshwater flux causes a density loss which decreases with decreasing $pCO_2$ as the atmosphere can hold less moisture with decreasing air temperature. As a result, the net density gain over the shelves increases with decreasing $pCO_2$.

Over the open-ocean convection area (Fig. 10b and c), there are two different modes, a thermal and a haline mode. The

thermal mode occurs in LGM-353 and LGM-284. For this mode, the $OOC_{min}$ case best describes the surface density flux. There is a net density gain caused by heat loss. The brine component is negative, reducing the density gain through net sea-ice melt. Because the considered area in the $OOC_{max}$ case is larger than the actual open-ocean convection area, the sea-ice melt dominates the net density flux, resulting in an overall density loss in the $OOC_{max}$ case. The haline mode occurs in LGM-ref and LGM-149. Here, the density gain is dominated by brine release. As over the shelves, the heat-flux component becomes

small because the water temperature is already close to the freezing point. This is true for both the $OOC_{max}$ and the $OOC_{min}$ case. The $OOC_{min}$ case, however, underestimates the effect of the brine release because the considered area is smaller than the actual open-ocean convection area. The balance in LGM-230 lies in between the two modes, the brine component is close to zero and the heat-flux component and the freshwater component add to a small density loss.

The annual mean density at 500 m averaged over the respective regions shows that the shelf water is lighter than the open-

ocean convection water in LGM-353 and LGM-284 (not shown). In LGM-230, the density of the two water masses is very similar, and in LGM-ref and LGM-149, the shelf waters become denser than the open-ocean convection water. So only in these two experiments can the very salty waters from the shelves reach deeper layers and contribute to AABW formation. Hence, we conclude that the contribution of Weddell Sea shelf waters to AABW is key to the shoaling of the NADW cell in response to the GHG reduction in MPI-ESM.

## 6.5 The effect of brine release

Only those PMIP2 models which simulated a north-south salinity gradient similar to the one reconstructed by Adkins et al. (2002) also simulated a shallower NADW cell under glacial conditions. These models had a strong haline contribution to the surface density flux in the Southern Ocean, driven by sea-ice formation and brine release (Otto-Bliesner et al., 2007). The importance of brine release for the glacial overturning state in a CCSM3 simulation had already been identified by Shin et al.

(2003). However, so far no study has tried to quantify the effect of brine release on the glacial overturning. In order to estimate the effect of brine release for the AABW formation and overturning strength in our simulations, we perform three additional sensitivity experiments in which brine release on the Southern Hemisphere is reduced. To this end, we set the salinity of sea ice to $20\,\mathrm{g\,kg^{-1}}$ on the Southern Hemisphere (instead of $5\,\mathrm{g\,kg^{-1}}$), thus reducing the amount of brine that is released when sea ice is formed in the Southern Ocean by roughly 50 %. We apply these changes to piTOPO, LGM-ref and LGM-149 (see piTOPO-



brine, LGM-ref-brine and LGM-149-brine in Table 1) and analyse the effect of the reduced brine release on the surface density flux, the Atlantic hydrography and the overturning.

### 6.5.1 Surface density flux changes

Shelf convection is weakened in all three sensitivity experiments because the net density gain is reduced as a direct con-
sequence of the reduced brine release (see open symbols in Fig. 10). Open-ocean convection is reduced in the two glacial sensitivity experiments LGM-ref-brine and LGM-149-brine. In piTOPO-brine, however, the heat-flux component increases, and this increase results in enhanced open-ocean convection with respect to piTOPO.

### 6.5.2 Atlantic hydrography changes

The resulting temperature and salinity differences between piTOPO-brine and piTOPO are relatively small(Fig. 11a and d).
The most prominent temperature signal is a northward shift of the Antarctic Circumpolar Current front. The northward shift is consistent with an increased open-ocean convection area in the Weddell Sea. The largest salinity change is also associated with the northward shift of the Antarctic Circumpolar Current front. A weak freshening of AABW occurs below 3000 m in the North Atlantic.

In LGM-ref-brine and LGM-149-brine, the resulting temperature and salinity changes are much larger than in piTOPO-
brine. In both experiments, there is a warming of up to 2.4°C below 3000 m (Fig. 11b and c), which indicates that NADW replaces AABW in the deep North Atlantic. The salinity differences show a freshening of AABW and a salinity increase of NADW in both experiments (Fig. 11d and f). The freshening of AABW and the salinity increase of NADW lead to a reduction of the density difference between the two water masses, as AABW becomes lighter and NADW becomes denser. Therefore, NADW can replace AABW in the deep North Atlantic.

### 6.5.3 Overturning changes

The response of the overturning reflects the hydrographic changes. Changes in overturning strength and geometry are very small from piTOPO to piTOPO-brine (Fig. 12). In both LGM-ref-brine and LGM-149-brine, the lighter AABW induces a weakening of the AABW cell and the NADW cell deepens and strengthens. In LGM-149-brine, the effect is strongest with a deepening of the NADW cell of 600 m and an increase in overturning strength of 2.5 Sv. These results confirm that the
contribution of the very salty coastal water to the formation of AABW is key to the shoaling of the NADW cell in LGM-ref and LGM-149 in MPI-ESM.

## 7 Discussion

Previous studies have identified both the Southern Ocean and the North Atlantic as the origin of the salinity increase and expansion of AABW and the subsequent shoaling of the NADW cell. A comparison of the water mass formation rates of
NADW and AABW in present-day and glacial simulations of CCSM3 suggested that the shoaling of the NADW cell in the



glacial simulation was caused by changes in the Southern Ocean haline density flux and not by changes in the North Atlantic density flux (Shin et al., 2003). The changes in the Southern Ocean haline density flux were attributed to the expansion of Antarctic sea ice and brine release (see also Ferrari et al., 2014). Simulations with a regional ocean model coupled to an ice-shelf-cavity model, on the other hand, suggested that the cooling of NADW is the driver of the salinity increase of AABW

(Miller et al., 2012; Adkins, 2013). Colder NADW would decrease the basal melting of the Antarctic ice sheet and thus increase the salinity of AABW. Miller et al. (2012) stated that the effect of increased brine release due to ocean cooling was negligible as it was compensated by decreased evaporation. It is, however, questionable whether the surface density flux in their experiments was representative for glacial conditions, because they used present-day forcing for the ocean model in all their experiments and cooled the water column directly at the open boundaries of the model domain. The changes of the haline density flux might

therefore be underestimated in their experiments. Because MPI-ESM does not account for basal ice-sheet melt, the reduced basal ice-sheet melting cannot be the driver in our simulations. Our sensitivity experiments confirm that changes in the haline density flux of the Southern Ocean are the main driver of the salinity increase of AABW and that brine release is a key factor for the shoaling of the NADW cell.

LGM-149 is the only experiment in which a shoaling of the NADW cell with respect to the present-day state occurs. Further

cooling than that induced by the glacial GHG concentrations is needed to dominate the deepening effect of the ice sheets. The shoaling takes place once the shelf-convection contribution to AABW becomes dominant. In piCTL, AABW is formed almost entirely through open-ocean convection, because the shelf waters are too light. This may be the result of missing shelf dynamics in the model and the simplified representation of ice-sheet mass loss. Therefore, the threshold beyond which changes in slope convection become relevant appears to be located at a too low $pCO_2$ in MPI-ESM. The location of the threshold might

depend on the way the ice-sheet runoff is treated in the model, because this has a direct effect on the freshwater flux (E-P+R). In MPI-ESM, the P-E over glaciers directly enters the runoff and is put into the ocean at the corresponding coastal grid point. This reduces the density immediately at the coast due to the additional freshwater input. The most realistic way to account for mass loss of the ice sheets would be to explicitly simulate the calving of ice bergs, which then melt at lower latitudes. A simpler way, which does not require an explicit ice-berg model, could be to put the ice-sheet runoff into the ocean at lower latitudes to

account for the melting of the icebergs. Stössel et al. (2015) showed that the properties of present-day AABW improved if the ice sheet runoff was distributed homogeneously over the Southern Ocean from the coast to 60°S instead of being put directly into the coastal grid points. This might also be true for the properties of glacial AABW.

In our experiments we study the sensitivity of the AMOC to different forcings in a setup with glacial ice sheets. It is, however, not a priori clear that the response to the different forcings and in particular to the GHG reduction is independent of the setup

of the reference state. The results of Oka et al. (2012) showed that the AMOC response to glacial cooling is quite sensitive to the applied wind stress and hence to the ice sheets. About twice the amount of cooling was needed to cross the thermal threshold in their experiments with glacial wind stress as compared to the experiments that used present-day wind stress (see their experiment series HT-CTL and HT-wind). In our LGM-*nnn* experiments we do not a find an abrupt change of the AMOC state as Oka et al. (2012) did in their simulations. The AMOC decrease is rather gradual, at least within the studied range of

GHG concentrations. The different response could be an effect of the mixed boundary conditions in the stand-alone ocean





model used by Oka et al. (2012) because mixed boundary conditions are known to cause an AMOC and deep convection that are overly sensitive to changes in forcing (e.g., Mikolajewicz and Maier-Reimer, 1994).

## 8 Conclusions

Based on our simulations with the coupled climate model MPI-ESM, we conclude the following:

1. The PMIP3 ice sheets induce a deepening of the NADW cell and an increase in the overturning strength caused by wind stress changes that favour NADW formation. The GHG reduction induces a shoaling of the NADW cell and a decrease in the overturning strength.

2. The effect of the ice sheets and the GHG reduction compensate each other in the deep ocean, so that no shoaling of the boundary between NADW and AABW is simulated in the glacial reference simulation LGM-ref.

3. Within the studied $pCO_2$ range, there is no threshold beyond which an abrupt decline of the overturning strength occurs in the glacial setup. Instead, we find a gradual decline.

4. Brine rejection in the Southern Ocean is key to the shoaling of the boundary between NADW and AABW. Shoaling sets in only below an atmospheric $pCO_2$ of 230 ppm, when Southern Ocean shelf water becomes denser than open-ocean convection water and contributes significantly to AABW formation.

5. The GHG concentrations needed to induce changes in the Southern Ocean shelf convection, which in turn result in a shoaling of the NADW cell and a reversal of the north-south salinity gradient with respect to piCTL, are too low in MPI-ESM. Therefore, the simulated state of the AMOC and the deep ocean in LGM-149 is closer to the reconstructed glacial state than that in LGM-ref.

*Acknowledgements.* This research was supported by the International Max Planck Research School on Earth System Modelling. All simu-
20  lations were performed at the German Climate Computing Center (DKRZ). The authors thank Thomas Kleinen for comments that improved
the manuscript and Florian Ziemen for helpful discussions.



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





**Table 1.** List of experiments and the respective forcing configurations. $pCO_2$ is given in ppm, $pN_2O$ and $pCH_4$ in ppb. The length of the simulations is given in years.

| Experiment | Orbit | Ice sheets | $pCO_2/pN_2O/pCH_4$ | Length | Other changes |
|---|---|---|---|---|---|
| piCTL | 0k | 0k | 284/791/275 | 1700 | |
| piTOPO | 21k | 0k | 284/791/275 | 3900 | |
| piTOPO-brine | 21k | 0k | 284/791/275 | 2200 | $S_{seaice,SO}$=20 g kg$^{-1}$ |
| LGM-353 | 21k | 21k | 353/1078/318 | 1400 | |
| LGM-284 | 21k | 21k | 284/791/275 | 2000 | |
| LGM-230 | 21k | 21k | 230/548/236 | 1400 | |
| LGM-ref | 21k | 21k | 185/350/200 | 2300 | |
| LGM-149 | 21k | 21k | 149/196/162 | 2800 | |
| LGM-ref-brine | 21k | 21k | 185/350/200 | 1400 | $S_{seaice,SO}$=20 g kg$^{-1}$ |
| LGM-149-brine | 21k | 21k | 149/196/162 | 1800 | $S_{seaice,SO}$=20 g kg$^{-1}$ |

**Table 2.** Orbital parameters for present day (0k) and LGM (21k).

| Orbit | Eccentricity | Perihelion | Obliquity |
|---|---|---|---|
| 0k | 0.0167724 | 282.04 | 23.446 |
| 21k | 0.018994 | 294.42 | 22.949 |





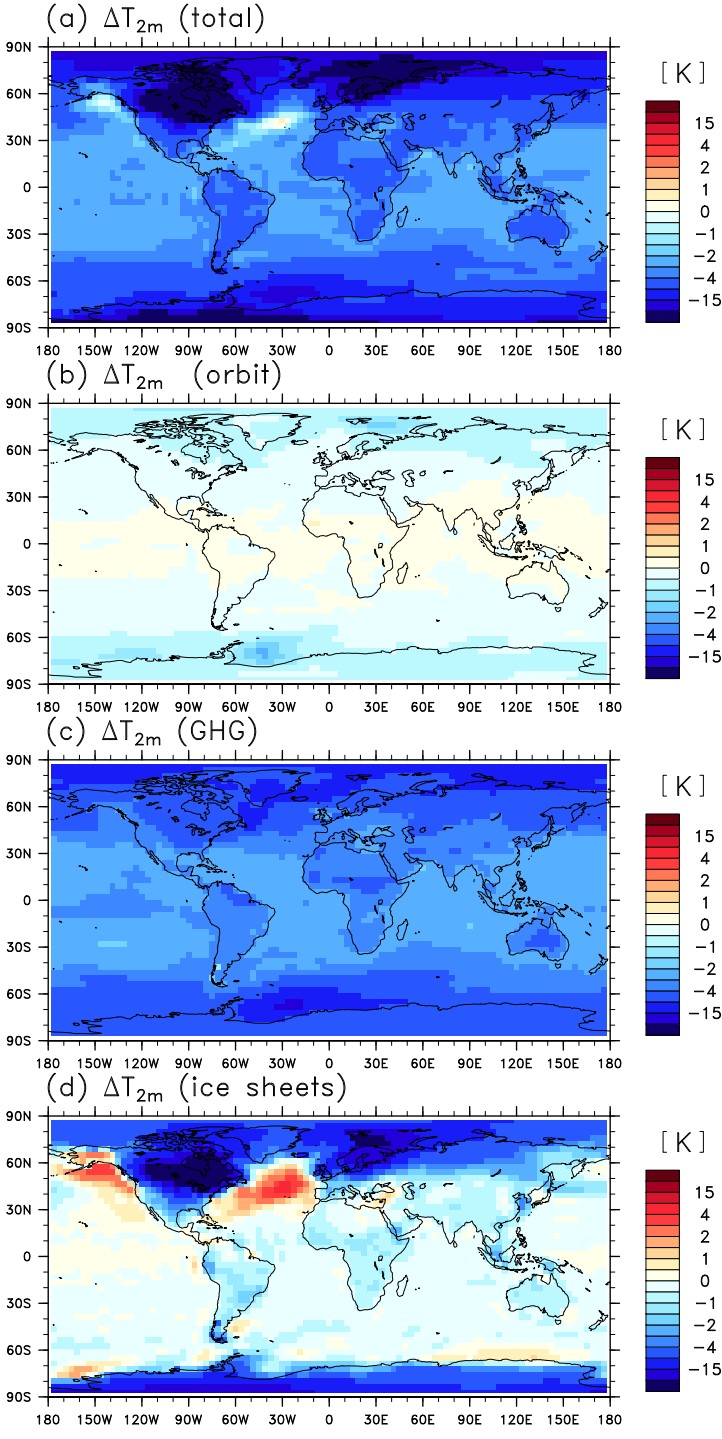

**Figure 1.** 2m air temperature differences in response to **(a)** the total glacial forcing (LGM-ref – piCTL), **(b)** the orbital configuration (piTOPO – piCTL), **(c)** the glacial GHG concentrations (LGM-ref – LGM-284) and **(d)** the ice sheets (LGM-284 – piTOPO).





**Figure 2.** Comparison of the temperature differences within the model to reconstructed temperature differences at the proxy sites (LGM-ref – piCTL). **(a)** Land based temperature differences from Bartlein et al. (2011) (dark green triangles), additional points from the compilation by Shakun et al. (2012) provided in the online supporting material of Schmittner et al. (2011) (light green circles) and simulated surface temperature differences (without height correction: grey stars, with height correction: black stars). **(b)** SST differences from MARGO (coloured symbols) and simulated SST differences (black stars).



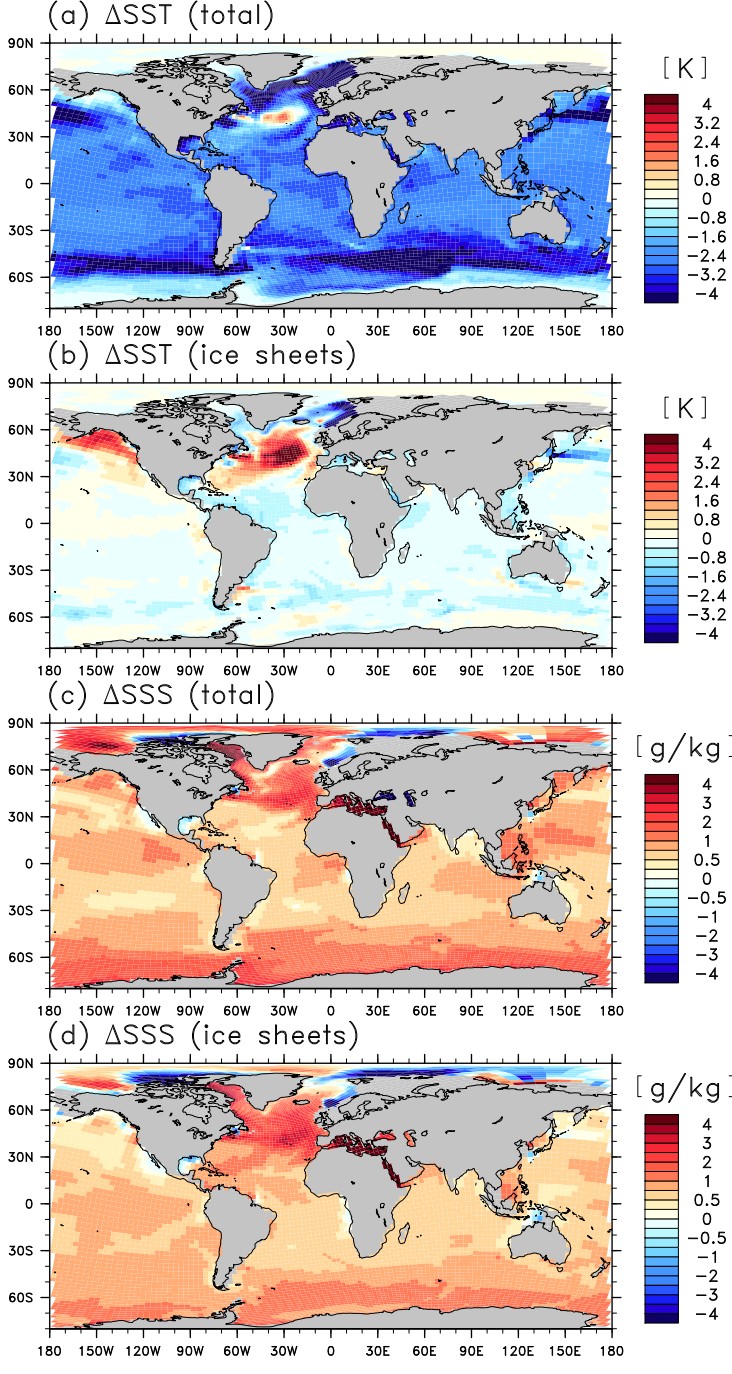

**Figure 3. (a)** SST response to the total glacial forcing (LGM-ref – piCTL) **(b)** SST response to the ice-sheets (LGM-ref – LGM-284), **(c)** sea surface salinity (SSS) response to the combined glacial forcings (LGM-ref – piCTL) and **(d)** SSS response to the ice sheets (LGM-ref – LGM-284).





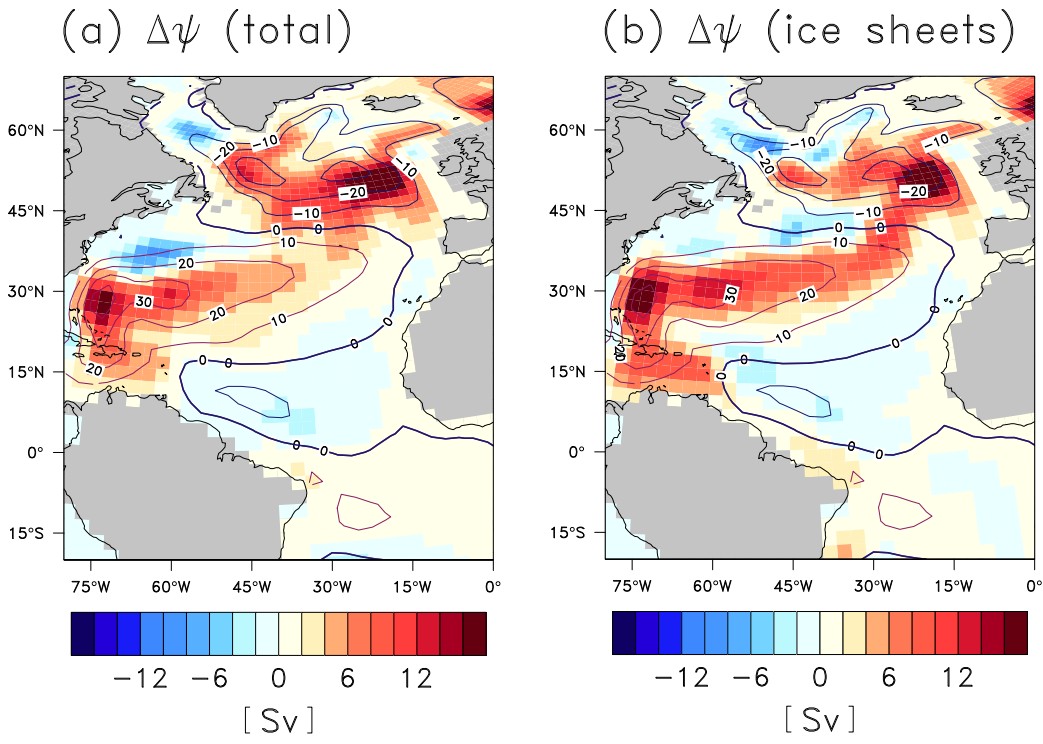

**Figure 4.** Changes in the barotropic stream function induced by **(a)** the total glacial forcing (LGM-ref – piCTL) and **(b)** the ice sheets (LGM-284 – piTOPO). Superimposed contours show the absolute barotropic stream function in the respective reference experiment, i.e., piCTL in (a) and piTOPO in (b). Cyclonic rotation is indicated by blue contours, anti-cyclonic rotation by red contours.





**Figure 5.** Transect through the Atlantic. **(a-d)** From top to bottom: potential temperature in piCTL and temperature changes due to the total glacial forcing (LGM-ref – piCTL), the glacial GHG concentrations (LGM-ref – LGM-284) and the ice sheets (LGM-284 – piTOPO). **(e-h)** As left column but for salinity. Before calculating the differences, the salinity of piCTL and piTOPO has been increased by 1.21 g kg$^{-1}$ for comparison with the GHG induced differences (LGM-ref – LGM-284). A map of the transect is shown in the top left panel.



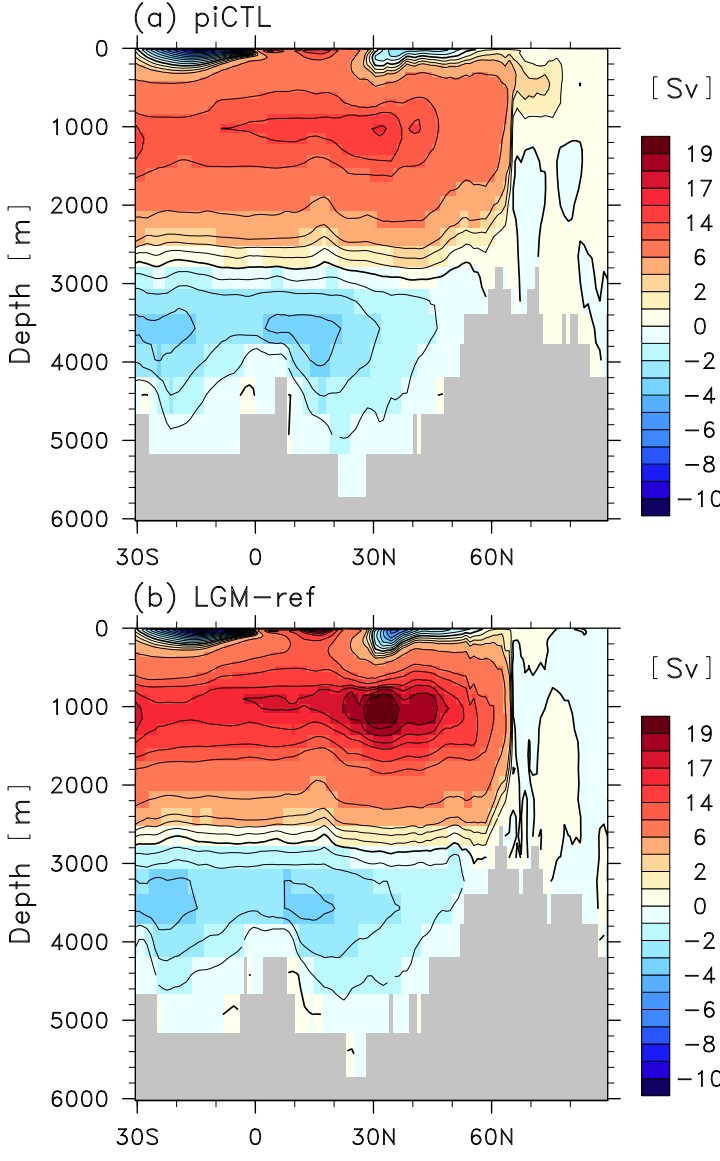

**Figure 6.** Atlantic meridional overturning stream function for **(a)** the preindustrial state and **(b)** the glacial state. Red shading indicates clockwise, blue shading anti-clockwise flow. Note that the contour levels are not symmetric.





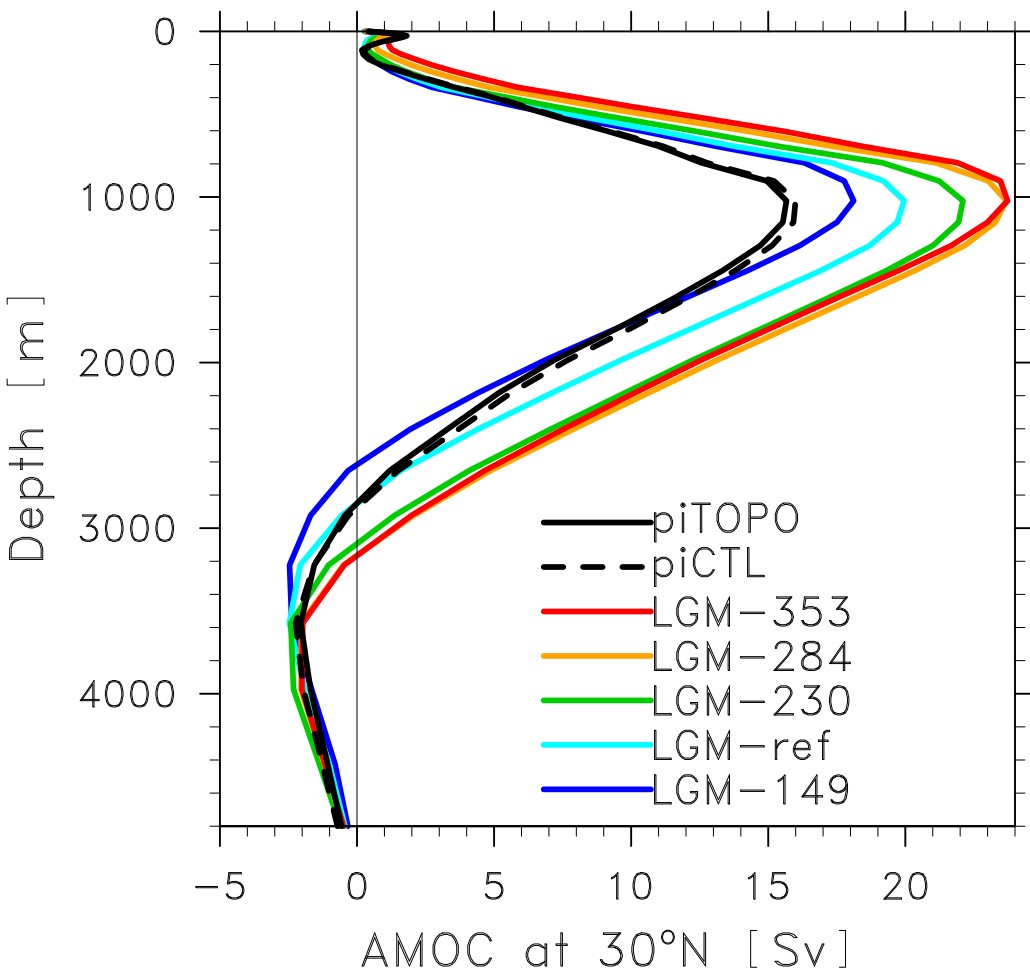

**Figure 7.** Profile of the AMOC at 30°N for the different experiments. Experiments with LGM setup are shown in color, experiments with piTOPO setup are shown in black and gray.



**Figure 8.** Climatological mean winter (JFM) mixed layer depth (shading), winter (JFM) ice edge (30 percent coverage, black contour) and summer ice edge (30 percent coverage, red contour).





**Figure 9.** Maximum of **(a)** $\sigma_\Theta$ and **(b)** $\sigma_2$ at 560 m in the Nordic Seas and at 2000 m in the North Atlantic and Weddell Sea as a function of $pCO_2$ in the glacial setup. **(c)** Temperature and salinity corresponding to the maximum density in each region. Solid contours indicate $\sigma_\Theta$, dotted contours indicate $\sigma_2$. The contour interval is 0.2 kg m$^{-3}$. For an easier comparison with the glacial setup runs, the salinity of piCTL and piTOPO has been offset by 1.21 g kg$^{-1}$ to account for the global mean salinity difference caused by the lower glacial sea level. The small map shows the definition of the three selected regions.





**Figure 10.** Components of the annual mean density flux integrated over different regions of the Weddell Sea as a function of $pCO_2$ integrated over **(a)** the shelf area defined by the 1000 m isobath, **(b)** the open-ocean convection area defined by the maximum extent of MLD>2000 m from all experiments and **(c)** the open-ocean convection area defined by the minimum extent of MLD>2000 m from all experiments. Positive values indicate a density gain, negative values a density loss. The x-axis is scaled logarithmically. Open symbols indicate the experiments with reduced brine release. Circles indicate the LGM-*nnn* experiments, the triangles represent the piTOPO experiments.





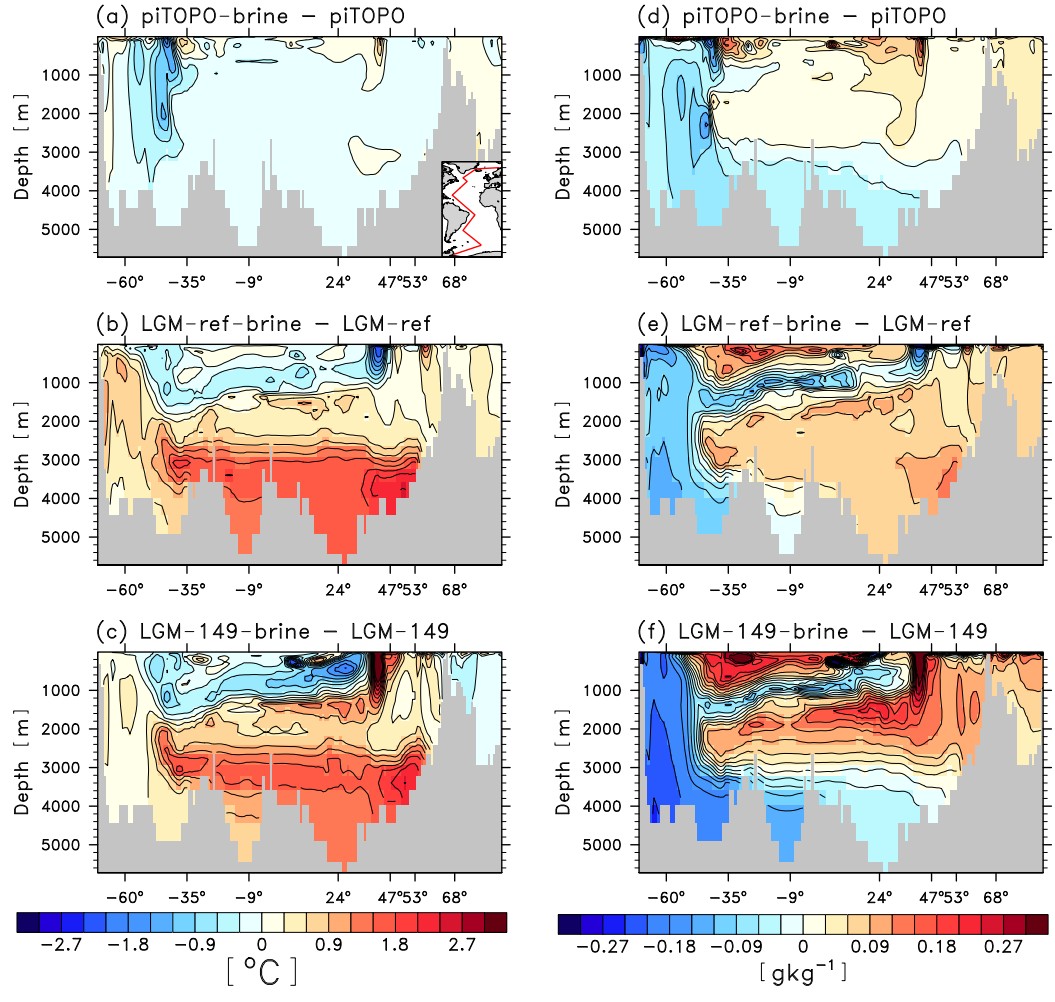

**Figure 11.** Temperature and salinity changes caused by the reduced brine release. Shown is the same transect through the Atlantic as in Fig. 5. Temperature difference for **(a)** piTOPO-brine – piTOPO, **(b)** LGM-ref-brine – LGM-ref and **(c)** LGM-149-brine – LGM-149. Salinity difference for **(d)** piTOPO-brine – piTOPO, **(e)** LGM-ref-brine – LGM-ref and **(f)** LGM-149-brine – LGM-149. Contour intervals are 0.3°C and 0.03 g kg$^{-1}$, respectively.





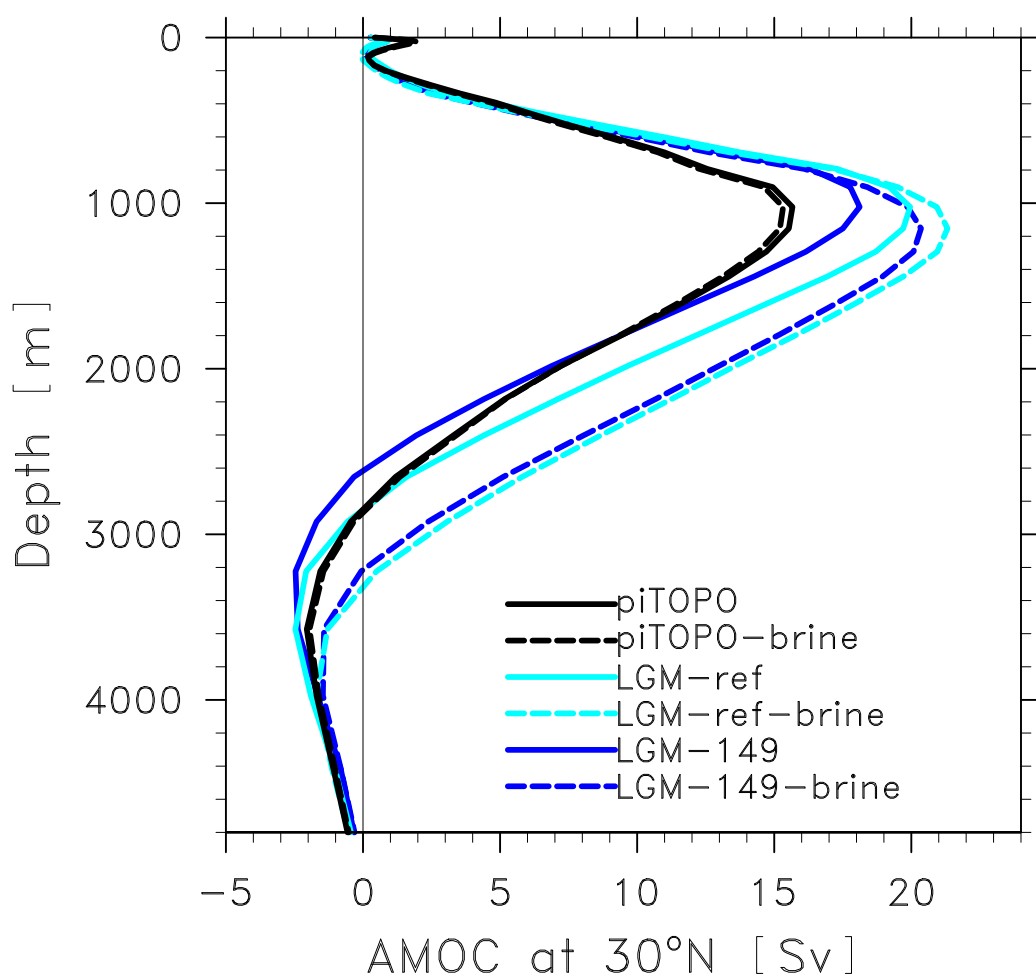

**Figure 12.** Profile of the AMOC at 30°N. Experiments with reduced brine release are indicated by dashed lines.