# Peer review of "The effect of greenhouse gas concentrations and ice sheets on the"

_Climate of the Past, 2016_

## Referee Comment (RC1) · Anonymous Referee #1 · 5 May 2016

This is an excellent paper. It discusses the role of ice sheets and atmospheric CO2 levels for the water mass distribution and meridional circulation characteristics of the LGM. It shows that the MPI model is able to produce a weaker and shallower AMOC as reconstructions suggest, but the atmospheric CO2 value used to produce it is too low.

The only comment that I am going to make on this paper is on the way comparisons are made. Instead of comparing different experiments (LGM GHG, LGM ice sheets, etc.) to a sole default run (e.g., a preindustrial run), they compare the LGM GHG experiment to an LGM run with preindustrial GHG to get GHG effects, and then compare the latter one to a preindustrial run with LGM orbital parameters to isolate ice sheet+topography

effects. Although correct, I think this is confusing. When looking at the pictures, I had to switch back and forth to the experiment design section because I didn't remember what was being compared with what in each of them. A way to solve this without having to restructure all the images would be to add a footnote in each panel specifying what is being compared.

Otherwise the paper should be published by Climate of the Past

———————————————————

---

## Referee Comment (RC2) · Anonymous Referee #2 · 20 Jun 2016

By using the coarse resolution version of the Max Plank Institute Earth System Model (MPI-ESM), the authors investigated the response of the AMOC and deep ocean water masses to glacial forcing. The authors' sensitivity simulations clearly demonstrated that the low greenhouse gas concentration contributes to causing the weaker AMOC and shoaling of NADW, whereas effects of the ice sheets tend to cause the stronger AMOC and deepening of NADW; balance of these two effects mainly controls the glacial response of the AMOC and NADW. Contrary to the proxy data, the authors' reference LGM simulation leads to the stronger AMOC and deepening of NADW. By applying additional cooling by setting 149ppm of $CO_2$ concentration, the authors obtained the AMOC closer to the proxy data. By performing the simulations under various level

of $CO_2$ concentrations (149, 185, 230, 284, 353ppm), the authors investigated the changes in the AMOC and water mass properties. The authors found that brine release over the Weddell Sea becomes dominant for determining surface density flux there in lower $CO_2$ (149, 185ppm) cases. The authors concluded that this brine release is important for a shoaling of glacial NADW, which was supported by their sensitivity simulations where the brine release is artificially reduced.

The manuscript is well-written and I think this is valuable contribution toward our understanding the glacial AMOC. Therefore, I would like to recommend the publication. Followings are several comments on the manuscript, which I hope will be helpful for the authors to prepare the final version of the manuscript.

Overall comments)

(1) The authors concluded that changes in convective system around the Weddell Sea, shifts from open convection to shelf convection, are important for shoaling of glacial AMOC. The changes in convection system in the Southern Ocean were not explicitly displayed in the manuscript, and addition of such figures might be important supporting evidence about the authors' conclusion.

(2) The authors obtained the shallower AMOC in their simulation LGM-149. However, even in LGM-149, the maximum value of the AMOC is still stronger than pi-CTL. This fact appears not directly mentioned and discussed in the manuscript, but explicit statement on this fact and discussion on it might be valuable.

(3) In the authors' model simulations, Fig. 8 (convection around the North Atlantic) suggests that the response of convection in the Labrador Sea is somewhat complicated: no Labrador Sea convection in pi-CTL → active convection in LGM-353 and LGM-284 → decreased convection in LGM-230 and LGM-ref → active again with shifted location in LGM-149. Considering the fact that the authors' model failed to reproduce the Labrador Sea convection in pi-CTL due to coarse resolution, I feel the possibility that this bias might affect the glacial response of the AMOC. Discussion on the role of the

Labrador Sea convection might be an additional important viewpoint for understanding the glacial response of the AMOC in the authors' model simulations.

Specific comments)

P4.L5: Why ICE5G is used for land-sea mask instead of that of PMIP3 ice sheet?

P4.L11: Please explicitly state what the letter "TOPO" stands for, or rename the experiment name (TOPO appears to remind us of topography effect and might be a little bit confusing).

P11.L24: What do the authors mean by "the last wet layer"?

P14.L10-11: The authors state that "the northward shift is consistent with an increased open-ocean convection". Would the authors explain explicitly what do they mean by "consistent"?

P15.L16: Here, the authors concluded that "the shoaling takes place once the shelf-convection contribution to AABW becomes dominant". I think this is one of the most important conclusions of the manuscript. Although the Figure 10 indicates that brine release is actually important for determining the surface density flux over the shelf regions, changes in convective system (i.e. shifts from open convection to shelf convection) are not explicitly displayed in the manuscript. I suggest the authors to add the figures which display changes in convective system in the Southern Ocean (also see my overall comment 3).

Fig.6: I think that addition of the AMOC figure in other simulations (LGM-353, LGM-284, LGM-230, LGM-149) will be meaningful information for the readers, although I understand that shoaling of the AMOC in LGM-149 can be confirmed from Fig.7.

---

## Author Comment (AC1) · 18 Jul 2016

We would like to thank the two anonymous reviewers for their comments and suggestions. Below is our point-by-point reply to the comments of the referees. Referee comments are written in italic font, our reply is written in normal font. Suggestions for reformulations are written in blue.

Reply to Anonymous Referee #1

Overall Comment: "*The only comment that I am going to make on this paper is on the way comparisons are made. Instead of comparing different experiments (LGM GHG, LGM ice sheets, etc.) to a sole default run (e.g., a preindustrial run), they compare the LGM GHG experiment to an LGM run with preindustrial GHG to get GHG effects, and then compare the latter one to a preindustrial run with LGM orbital parameters to isolate ice sheet+topography effects. Although correct, I think this is confusing. When looking at the pictures, I had to switch back and forth to the experiment design section because I didn't remember what was being compared with what in each of them. A way to solve this without having to restructure all the images would be to add a footnote in each panel specifying what is being compared*"

Reply: The above comment refers to Figure 1 and Figure 3-5. In order to make it easier to see immediately which experiments are being compared in which panel, we will move the name of the effect (i.e. "total", "GHG" and "ice sheets") into the figure and put the experiment names as a title. This way, the all information can be found both in each of the panels and in the figure caption.

In order to make the experiment names easier to read, we would also remove the hyphens in the experiment names to avoid confusion between hyphen and minus. We will also rename piTOPO to piORB (see reply to AR2's comment S2).

Example:

[Figure]

Reply to Anonymous Referee #2

Overall Comments:

O1: "*The authors concluded that changes in convective system around the Weddell Sea, shifts from open convection to shelf convection, are important for shoaling of glacial AMOC. The changes in convection system in the Southern Ocean were not explicitly displayed in the manuscript, and addition of such figures might be important supporting evidence about the authors' conclusion.*"

Reply: Our conclusion was based on a TS-diagram (see Figure O1), in which we compared the water mass properties of the open-ocean convection area and over the shelves of the Weddell Sea. This TS-diagram is discussed in the manuscript on page 13, lines 19-21. It is, however, not shown. We propose to add the TS-diagram to the manuscript as support for our conclusion about the importance of shelf convection.

[Figure]

Figure O1: TS-diagram similar to Fig. 9c in the manuscript. Shown are the mean temperature and salinity over the Weddell Sea shelves (diamonds) and over the $OOC_{max}$ region (circles). Open symbols indicate the experiments with reduced brine release. Solid contours indicate $\sigma_\Theta$, dotted contours indicate $\sigma_2$.

O2: "*The authors obtained the shallower AMOC in their simulation LGM-149. However, even in LGM-149, the maximum value of the AMOC is still stronger than pi-CTL. This fact appears not directly mentioned and discussed in the manuscript, but explicit statement on this fact and discussion on it might be valuable.*"

Reply: This can be easily added. We have focused here only on the shoaling because it is the most reliable information which can be obtained from the reconstructions. The strength is more difficult to constrain. Recent studies by Lippold et al (2012) and Böhm et al (2015) have shown that also an AMOC stronger than or at least as strong as today can be inferred from the proxies. Nonetheless, we agree that it is worth mentioning and discussing this fact at the end of chapter 6.1.

O3: "*In the authors' model simulations, Fig. 8 (convection around the North Atlantic) suggests that the response of convection in the Labrador Sea is somewhat complicated: no Labrador Sea convection in pi-CTL -> active convection in LGM-353 and LGM-284 -> decreased convection in LGM-230 and LGM-ref -> active again with shifted location in LGM-149. Considering the fact that the authors' model failed to reproduce the Labrador Sea convection in pi-CTL due to coarse resolution, I feel the possibility that this bias might affect the glacial response of the AMOC. Discussion on the role of the Labrador Sea convection might be an additional important viewpoint for understanding the glacial response of the AMOC in the authors' model simulations.*"

Reply: The above comment made us revisit the Labrador Sea convection in the piCTL simulation. We looked at the yearly maximum of the mixed layer depth (MLD) in the Labrador Sea during the 300 years of which we calculated the average (see Figure below). The timeseries shows that the model does actually not fail to capture Labrador Sea convection in piCTL, there are many years with deep convection with MLDs down to 3500. The deep convection seems to vary on pentadal to decadal time scales. In the years without deep convection, the MLDs are very shallow. The very shallow MLDs of 400 m to 600 m in Figure 8 are therefore the result of two effects; deep convection does not take place every year and it varies spatially. We would therefore propose to change the sentences on p.11, l.8-10 and rewrite the description of the Labrador Sea convection in piCTL.

[Figure]

Figure O3: Time series of the yearly maximum mixed layer depth occurring in the Labrador Sea in the piCTL simulation.

As stated on p.11,l.10, 'higher-resolution versions of MPI-ESM show MLDs down to 3000 m in the Labrador Sea'. With the higher resolution, Labrador Sea convection takes place every year; therefore it is more visible in the long-term average. We have compared the total glacial response of the AMOC in the higher-resolution CMIP5/PMIP3 experiments with our findings to see, whether the continuous Labrador Sea convection might have an effect on the total response. In both the high-resolution piCTL and the high-resolution LGM simulation the AMOC is somewhat stronger than in our coarse-resolution experiments and the maximum AMOC strength shifts from 30N to 35N. But the total glacial response is the same. There is no change in the vertical extent of the NADW cell and the maximum AMOC strength increases by about 4 Sv. Therefore, we conclude that the glacial response is independent of the model resolution and does not depend on whether the Labrador Sea convection is intermittent or continuous.

Suggestion for first paragraph of Chapter 6.2 (p.11,l.7-11): In both piCTL and piORB, NADW formation through deep convection takes place mainly in the ice-free part of the Nordic Seas (Fig.8a, only piCTL is shown). In the Labrador Sea, deep convection varies on pentadal to decadal time scales with years in which MLDs go as deep as 3400 m and years where no deep convection occurs. In addition, the exact location of the deep convection varies in time.  The long-term mean MLDs in the Labrador Sea are therefore rather shallow with 400 m to 600 m. Higher-resolution versions of MPI-ESM simulate continuous deep convection in the Labrador Sea (see e.g., Jungclaus et al, 2013). The different behavior of the Labrador Sea convection with resolution does not affect the total glacial response of the AMOC. Comparing the preindustrial control simulation and the LGM simulation of MPI-ESM-P in the CMIP5/PMIP3 database shows that the depth of the NADW cell remains almost unchanged and the maximum overturning strength increases (see also Table 1 in Muglia and Schmittner, 2015).

Specific comments:

S1: "*Why ICE5G is used for land-sea mask instead that of PMIP3 ice sheets?*"

Reply: This has practical reasons. In MPI-ESM, the land-sea mask is defined by the ocean bathymetry, and since we used ICE5G for the ocean bathymetry, also the land-sea mask is taken from ICE5G.

S2: "*Please explicitly state what the letter "TOPO" stands for, or rename the experiment name (TOPO appears to remind us of topography effect and might be a little bit confusing).*"

Reply: 'TOPO' was supposed to indicate that we are using preindustrial ice sheets and topography. But it is true that this choice of name is somewhat misleading. We would therefore propose to rename the experiment to piORB, where the 'pi' would indicate preindustrial conditions and 'ORB' would indicate the changed orbit.

S3: "*What do the authors mean by "the last wet layer"?*"

Reply: This refers to the glacial ocean-model bathymetry. In the model, the Iceland-Scotland-Channel is 19 model levels deep and the depth of last model level which is still ocean, i.e. "wet", corresponds to 560 m. We suggest the following reformulation:.

Suggestion for p.11,l.23-25: In the Nordic Seas, we determine the water-mass properties at 560 m depth which corresponds to depth of the deepest model layer with a connection between the glacial Nordic Seas and the glacial North East Atlantic.

S4:" *The authors state that "the northward shift is consistent with an increased open-ocean convection". Would the authors explain explicitly what do they mean by "consistent"?*"

Reply: In the piTOPO-brine experiment, the open-ocean convection area in the Weddell Sea expands towards the north and therefore also the ACC is shifted towards the north. We propose to reformulate the sentence for clarity.

Suggestion for p.14,l.10-11: The northward shift of the Antarctic Circumpolar Current front is associated with an expanded open-ocean convection area in the Weddell Sea.

S5: "*Here, the authors concluded that "the shoaling takes place once the shelf-convection contribution to AABW becomes dominant". I think this is one of the most important conclusions of the manuscript. Although the Figure 10 indicates that brine release is actually important for determining the surface density flux over the shelf regions, changes in convective system (i.e. shifts from open convection to shelf convection) are not explicitly displayed in the manuscript. I suggest the authors to add the figures which display changes in convective system in the Southern Ocean (also see my overall comment 3).*"

Reply:  See reply to O1

S6: "*I think that addition of the AMOC figure in other simulations (LGM-353, LGM-284, LGM-230, LGM-149) will be meaningful information for the readers, although I understand that shoaling of the AMOC in LGM-149 can be confirmed from Fig.7.*"

Reply: We chose not to show the 2D view of the AMOC for all experiments, because they take a lot of space and we thought that they would add little extra value. The most important information, i.e. the change in NADW cell depth and the change in maximum overturning strength is seen easiest in the 1D profile in Figure 7. If considered as absolutely essential, the 2D view could be added in a supplement:

[Figure]

References

Böhm, E., Lippold, J., Gutjahr, M., Frank, M., Blaser, P., Antz, B., Fohlmeister, J., Frank, N., Andersen, A., and Deininger, M.: Strong and deep Atlantic meridional overturning circulation during the last glcial cycle, Nature, 517, 73-76, 2015.

Jungclaus, J., Fischer, N., Haak, H., Lohmann, K., Marotzke, J., Matei, D., Mikolajewicz, U., Notz, D., and von Storch J.: Characteristics of the ocean simulations in the Max Planck Institute Ocean Model (MPIOM) the ocean component of the MPI-Earth system model, JAMES, 5, 422-446, 2013.

Lippold, J., Luo, Y., Francois, R., Allen, S. e., Gherardi, J. Pichat, S. Hickey, B., and Schulz, H.: Strength and geometry of the glacial Atlantic meridional Overturning Circulation, Nature geoscience, 5, 813-816, 2012.

Muglia, J. and Schmittner, A.: A Glacial Atlantic overturning increased by wind stress in climate models, Geophysical research Letters, 41, 2015.

---

## Author Response (AR1)

Hamburg, 28.07.2016

Dear Editor,

We have changed the manuscript according to our suggestions in the final author response. The point-by-point reply to the reviewers comments was already given in the final authors response, therefore we do not repeat it here again. The changes can be tracked in the marked-up version of the manuscript which can be found below. In the marked-up version, old text is marked in red and crossed out; new text is marked in blue.

We have made a few minor modifications, which do not change the content of the manuscript but hopefully give additional clarity. They are tracked in the marked-up version of the manuscript and also listed below.

Best regards,

Marlene Klockmann on behalf of all co-authors

List of additional changes

Entire manuscript: As already proposed in the final author response, we have removed the hyphens from the experiment names in LGM-353,LGM-284,LGM-230,LGM-ref and LGM-149.

p.3,l.19-20: We have added a sentence on the river routing in the glacial setup for completeness.

p.4,l.1-3: We replaced '*very similar to the model version*' by '*the coarse-resolution equivalent of*' and added '*with dynamical vegetation*'. This describes the difference between our model setup and MPI-ESM-P more correctly. We have also replaced '*the same MPIOM version with higher resolution (MPI-ESM-LR and MPI-ESM-MR)*' by '*the MPIOM version used in CMIP5*' for clarity.

p.4,l.17: We have added the word '*individual*' before '*effect of ice sheets and GHG concentration*' to emphasise that we analyse the effects separately.

p.7,l.16: We have added '*at 26N*' behind '*RAPID-MOCHA array*' for completeness.

p.9,l.29-30: We have replaced the sentence: '*Surface cooling in the North Atlantic is outweighed by the surface freshening and NADW formation is reduced.*' by '*The density difference between the two water masses increases due to salinity changes.*' On long time scales, the density difference between NADW and AABW is more relevant for the AMOC than surface density anomalies.

p.9,l.31: We have replaced '*sea-ice expansion*' by '*enhanced sea-ice formation*', because it is the formation rather than the expansion of sea ice which increases the brine release.

p.12,l.29: We have replaced '*60 to 90S*' by '*south of 60S*' for better readability.

p.12,l.30: We have inserted '*atmospheric*' before '*freshwater fluxes*' to be more precise.

p.13,l2: We have inserted '*maximum*' before '*MLDs*' to be more precise.

p.13,l.18-27: We have exchanged the word '*mode*' with '*regime*'. This word choice seems more appropriate to us.

p.14,l.14: We have inserted '*As expected*' at the beginning of the sentence, because the reduction of shelf convection as a consequence of the reduced brine release is exactly what we intended to achieve with the sensitivity experiments.

p.14,l.17-19: We have added two sentences to explain more specifically why open-ocean convection in piORB-brine is enhanced with respect to piORB.

p.15,l.26: We have replaced '*dominant*' with '*relevant*'. This word choice seems more appropriate.

p.15,l.27: We have inserted '*too fresh and*' before '*too light*' and added the reference to the new TS-diagram in Fig. 11.

p.29, Figure 7: We have corrected the figure caption.

[revised manuscript text omitted]